# Clinical Characteristics and Prognosis of HER2-0 and HER2-Low-Positive Breast Cancer Patients: Real-World Data from Patients Treated with Neoadjuvant Chemotherapy

**DOI:** 10.3390/cancers15194678

**Published:** 2023-09-22

**Authors:** Patrik Pöschke, Peter A. Fasching, Werner Adler, Matthias Rübner, Matthias W. Beckmann, Carolin C. Hack, Felix Heindl, Arndt Hartmann, Ramona Erber, Paul Gass

**Affiliations:** 1Department of Gynecology and Obstetrics, Erlangen University Hospital, Comprehensive Cancer Center Erlangen-EMN (CCC ER-EMN), Friedrich-Alexander–Universität Erlangen-Nürnberg (FAU), 91054 Erlangen, Germany; patrik.poeschke@uk-erlangen.de (P.P.); peter.fasching@fau.de (P.A.F.); matthias.ruebner@uk-erlangen.de (M.R.); matthias.beckmann@uk-erlangen.de (M.W.B.); carolin.hack@uk-erlangen.de (C.C.H.); felix.heindl@uk-erlangen.de (F.H.); 2Department of Medical Informatics, Biometry and Epidemiology, Friedrich-Alexander–Universität Erlangen-Nürnberg (FAU), 91054 Erlangen, Germany; werner.adler@fau.de; 3Institute of Pathology, Erlangen University Hospital, Comprehensive Cancer Center Erlangen-EMN (CCC-ER-EMN), Friedrich-Alexander–Universität Erlangen-Nürnberg (FAU), 91054 Erlangen, Germany; arndt.hartmann@uk-erlangen.de (A.H.); ramona.erber@uk-erlangen.de (R.E.)

**Keywords:** breast cancer, neoadjuvant therapy, prognosis, HER2-low-positive, HER2-0, HER2

## Abstract

**Simple Summary:**

A new subgroup of breast cancer patients with different survival outcomes, even before new treatment options have been approved, which can be easily differentiated immunohistochemically, has recently received great attention. The approval of trastuzumab deruxtecan in August 2022 by the Food and Drug Administration (FDA) for patients with HER2-low-positive metastatic pretreated breast cancer suggested a new breast cancer subtype; this implied new cohorts of patients with different overall survival outcomes and therefore different treatment needs and chances. Therefore, in our study, we examined whether we could retrospectively differentiate HER-0 and HER2-low-positive breast cancer patients treated with neoadjuvant chemotherapy in a long-term follow-up over 20 years. We observed the overall survival, disease-free survival, and pathological complete response rate.

**Abstract:**

In our study, we observed the long-term survival outcomes investigated for HER2-0 and HER2-low-positive breast cancer patients who received neoadjuvant chemotherapy. Between 1998 and 2020, 10,333 patients with primary breast cancer were treated, including 1373 patients with HER2-0 or HER2-low-positive disease with neoadjuvant chemotherapy. Descriptive analyses were performed, and logistic regression models and survival analyses were calculated for disease-free survival (DFS) and overall survival (OS). Among the 1373 patients, 930 (67.73%) had HER2-low-positive and 443 (32.27%) had HER2-0 tumors. Patients with HER2-0 tumors had a significantly better pathological complete response, 29.25% vs. 20.09%, and pathological complete response/in situ, 31.97% vs. 24.08%, than patients with HER2-low-positive tumors (*p* < 0.001; *p* = 0.003), regardless of the hormone receptor (HR) status. No statistically significant differences were observed for the HR-positive (*p* = 0.315; *p* = 0.43) or HR-negative subgroups (*p* = 0.573; *p* = 0.931). DFS and OS were significantly longer for HR-positive, HER2-low-positive patients (log-rank *p* = 0.02; *p* = 0.012). OS was significantly longer for HR-negative, HER2-0 patients (log-rank *p* = 0.032). No significant DFS differences were found for the HR-negative cohort (log-rank *p* = 0.232). For the overall cohort, no significant differences were noted between HER2-low-positive and HER2-0 patients, either for DFS (log-rank *p* = 0.220) or OS (log-rank *p* = 0.403). These results show different survival outcomes for HER2-0 and HER2-low-positive tumors relative to HR status. These different cohorts can be identified using standardized immunohistochemistry, even retrospectively.

## 1. Introduction

Breast cancer is the most common type of cancer in women [1]. The prognosis for breast cancer patients varies depending on many different factors, including tumor biology, tumor size, age at diagnosis, and body mass index. Human epidermal growth factor receptor 2 (HER2), as a prognostic and predictive factor, is overexpressed as a result of the amplification of the *ERBB2* gene, part of the epidermal growth receptor factor (EGFR) family, in about 15% of breast cancers [2,3]. The American Society of Clinical Oncology/College of American Pathologists (ASCO/CAP) guidelines recommend using immunohistochemistry (IHC) and in situ hybridization (ISH) to identify HER2-positive lesions [4]. IHC scores rank HER2 expression from 1+ to 3+, with a focus on overexpressing tumors categorized as an IHC score of 3+ or an IHC score of 2+ with positive ISH findings [4]. About 50% of breast cancer patients have HER2-low-positive tumors, about 80% of which represent hormone receptor (HR)-positive disease and about 20% of which involve triple-negative breast cancer (TNBC) [5,6,7]. Even an ESMO expert consensus statement (ECS) was published recently to understand the management of HER2-low-positive breast cancer [8].

HER2-positive tumors have recently become a target for antibody or antibody drug conjugate (ADC) therapies, as well as tyrosine kinase inhibitors [9].

HER2-low-positive tumors came into clinical consideration as a new category of breast cancer when fam-trastuzumab deruxtecan-nxki (T-DXd), a novel ADC, was found to be associated with significant progression-free survival in comparison with the physician’s choice of chemotherapy in patients with pretreated metastatic HER2-low-positive breast cancer [10,11]. A new tumor category has therefore emerged due to new treatment options, while at present, treatment decision-making is dominated by the HR status [12], and traditional anti-HER2 therapies such as trastuzumab have not shown any significant improvements in survival in HER2-low-positive breast cancer patients [13]. Even at the moment, however, HER2-low-positive tumors appear to be associated with longer overall survival in node-negative breast cancer patients and patients treated with neoadjuvant chemotherapy without any of the new treatment options [14,15].

The aim of the present analysis was therefore to assess whether there are any differences in survival for HER2-low and HER2-0 patients in relation to expected changes in the range of available treatments for HER2-low patients. The study involved a long-term follow-up period with real-world data from patients treated with neoadjuvant chemotherapy.

## 2. Materials and Methods

### 2.1. Patient Selection

A total of 10,333 patients with primary breast cancer were treated at the University Breast Center for Franconia in Erlangen University Hospital from the end of 1998 to the end of 2020. To ensure full documentation of local recurrences, distant metastases, and deaths, patients were contacted once a year unless the follow-up had already been carried out at the University Hospital’s breast center.

The eligibility of patients for inclusion in the study was defined by the following criteria. Excluded from the study, among the 10,333 patients who were at least 18 years of age and had been diagnosed with invasive breast cancer were male patients, those with bilateral breast or metastatic breast cancer, patients with events during the first 30 days after initial diagnosis, patients with missing HER2 IHC data, and HER2-positive (categorized as IHC score of 3+ or IHC score of 2+ with positive ISH findings) patients. The final sample consisted of 1373 patients (Figure 1). The ethics committee of Erlangen University Hospital provided approval for the study (ref. numbers 2700 and 297 17 Bc).

### 2.2. Histopathological Data

All histopathological information used in this analysis was directly documented from the original pathological reports, which were reviewed by two investigators. Grading, tumor type, hormone receptor (HR) status (estrogen and progesterone receptor status, HER2 status, and proliferation status as assessed by Ki-67 staining) have been routinely recorded at the Breast Center since 1995. Pathological complete response (pCR), representing the complete disappearance of all invasive carcinoma cells in the breast and axillary lymph nodes (ypT0/ypN0), was assessed pathologically in the resected tissue after neoadjuvant chemotherapy, and pathological complete response/in situ (pCR/is), representing an absence of invasive cancer in the breast and axillary nodes, irrespective of ductal carcinoma in situ, were also assessed [16,17]. In clinical routine work, the pretreatment core biopsies were stained as follows: monoclonal mouse/rat antibodies against Estrogen Receptor (ER)-alpha (before 2014: clone 1D5 1:200 dilution; since 2014: clone EP1, 1:40 dilution; DAKO, Glostrup, Denmark), monoclonal mouse antibody against the Progesterone Receptor (PgR) (before November 2018: clone pgR636, 1:200 dilution, DAKO, Denmark; since November 2018: PR, clone 1E2, Ventana, ready to use), and monoclonal antibody against Ki-67 (clone MIB-1, 1:200 dilution before 2015 and 1:100 dilution since 2015, DAKO, Denmark) were used to stain the preoperative core biopsies. The percentage of positively stained tumor cells was mentioned in the pathology reports. A polyclonal antibody against HER2 (1:200 dilution before 2012 and 1:1000 dilution since 2012, DAKO, Denmark) was used, and HER2 status was documented in the pathology reports as negative, 0, 1+, 2+, or 3+ in accordance with the published guidelines [18]. Tumors with a score of 0 or 1+ were defined as HER2-negative, and those with a score of 3+ were regarded as HER2-positive. Tumors with a 2+ staining were tested for gene copy numbers of HER2 using chromogene in situ hybridization. The gene copy numbers of HER2 and the centromeres of the corresponding chromosome 17 were stained using a kit with two probes of different colors (ZytoDot, 2C SPEC HER2/CEN17, Zyto Vision Ltd., Bremerhaven, Germany). A tumor was regarded as HER2-positive if the HER2/CEN17 ratio was ≥2.2 up to 2013 and ≥2 thereafter. The tumors were considered positive for the ER und PgR if at least 1% of cells were stained positive. HER2-low-positive status was defined as immunohistochemistry IHC 1+ or IHC 2+/in situ hybridization negative, and HER2-0 was defined as IHC 0, on the basis of the ASCO/CAP guidelines and the definition used in the literature for HER2-low-positive tumors, updated recently in a pathologist guideline update [4,10,11,14,15,19,20,21].

### 2.3. Clinical Data

The University Breast Center for Franconia is a breast cancer center certified by the German Society for Breast Diseases (Deutsche Gesellschaft für Senologie) and the German Cancer Society (Deutsche Krebsgesellschaft) [22,23]. During the annual audit process, treatment procedures are assessed, requiring treatment in accordance with the German guidelines for more than 95% of the patients. For data collection, epidemiological parameters and risk factors are correlated with a questionnaire completed by the patients. Briefly, all clinical and histopathological data were compiled prospectively in an annually audited, certified database.

### 2.4. Statistical Analysis

HER2-0 and HER2-low-positive patients were compared using logistic regression models in which age, body mass index (BMI), grading, HR status, tumor size, nodal status, and Ki-67 were used as independent variables in bivariate and multivariate models. These models were calculated for the subgroups of patients with (1) pCR yes; (2) pCR yes and HR-positive; (3) pCR yes and HR-negative; (4) pCR/is yes; (5) pCR/is yes and HR-positive; and (6) pCR/is yes and HR-negative. In each of the models, odds ratios (ORs) and their respective 95% confidence intervals were estimated for the predictor variables involved. To examine differences in overall survival (OS) and disease-free survival (DFS), Kaplan–Meier estimation was performed, and median survival, 10-year and 20-year survival rates, and their 95% confidence intervals were calculated for the several variables mentioned above. Follow-up times were censored at a maximum of 20 years. To examine differences in survival between HER2-0 and HER2-low-positive patients, Kaplan–Meier curves were produced and log-rank tests were calculated in the total group and in the HR-positive and HR-negative subgroups. Hazard ratios and their 95% confidence intervals were calculated using Cox proportional hazards models in bivariate and multivariate ways. Forest plots were created showing the odds ratios of HER2-low-positive versus HER2-0 patients for the risk of pCR and pCR/is, respectively, in the total patient collective and in the HR-positive and HR-negative subgroups. The significance level was set to 0.05. All statistical analyses were carried out using the R statistical package (version 4.2.2; R Development Core Team, Vienna, Austria, 2022) [24].

## 3. Results

Among the 1373 patients, 443 (32.3%) had HER2-0 tumors and 930 (67.7%) had HER2-low-positive tumors. The baseline characteristics for clinical and pathological parameters are shown in Table 1.

Patients with HER2-0 tumors had significantly higher pCR 29.25% vs. 20.09% and pCR/is rates of 31.97% vs. 24.08% than patients with HER2-low-positive tumors (*p* < 0.001 and *p* = 0.003), regardless of the HR status. No statistically significant differences were observed for HR-positive tumors (*p* = 0.315 and *p* = 0.43) or HR-negative tumors (*p* = 0.573 and *p* = 0.931) (Figure 2A,B).

In Table 2, pCR and pCR/is rates for different chemotherapy regimes are given, divided into HR positive and negative subgroups and HER2-0 and HER2-low-positive subgroups.

Similar results were seen in the univariate and multivariable logistic regression analyses (Figure 3).

Patients with HER2-low-positive and HR-positive tumors had significantly longer DFS periods (stratified log-rank test *p* = 0.02). No statistically significant differences were observed for the HR-negative cohort (stratified log-rank test *p* = 0.232) (Figure 4A–C). The DFS rates at 20 years for HR-positive patients were 43.2% (34.1–54.9%) for the HER2-0 cohort and 37.7% (20.5–69.3%) for the HER2-low-positive cohort. DFS rates at 20 years for HR-negative patients were not applicable for the HER2-0 cohort and 45.8% (33.9–61.9%) for the HER2-low-positive cohort; after 10 years, the DFS was 68.6% (62.1–77.0%) for the HER2-0 cohort and 61.5% (54.4–69.6%) for the HER2-low-positive cohort.

Patients with HER2-low-positive tumors had a statistically significantly longer OS in the HR-positive cohort (stratified log-rank test *p* = 0.012) and a statistically significantly shorter OS with HR-negative tumors (stratified log-rank test *p* = 0.032) (Figure 5A–C). The OS rate for HR-positive patients at 20 years was 51.0% (41.5–62.6%) for the HER2-0 cohort and 46.9% (26.2–83.8%) for the HER2-low-positive cohort. The OS rate for HR-negative patients at 10 years was 77.2% (70.3–84.8%) for the HER2-0 cohort and 67.2% (60.1–75.0%) for the HER2-low-positive cohort; after 20 years, it was not applicable for the HER2-0 cohort and 46.0% (33.3–63.6%) for the HER2-low-positive cohort.

In the overall patient group, no statistically significant differences in DFS were observed between HER2-low-positive and HER2-0 patients (log-rank *p* = 0.22), nor were significant differences seen in OS (log-rank *p* = 0.403) (Figure 5).

The results from the multivariate Cox regression model for the DFS and OS of HER2-0 vs. HER2-low-positive patients, divided into two groups of HR-positive and HR-negative patients, are shown in Table 3 and Table 4.

## 4. Discussion

This retrospective study analyzed a large cohort of patients treated with neoadjuvant chemotherapy at a single-center certified breast cancer center in Germany, with up to 20 years of follow-up, to evaluate the differences in survival parameters for breast cancer patients with HER2-0 and HER2-low-positive tumors. The main finding was a statistically significant difference in the overall and disease-free survival between these groups in both HR-positive and HR-negative patients, but not in the overall group.

Denkert et al. published a pooled analysis of patients treated in four different neoadjuvant trials in order to compare HER2-0 and HER2-low-positive tumors. HER2-low-positive, HR-positive tumors were found to be associated with a significantly lower pCR rate in comparison with HER2-0, as well as significantly longer 3-year DFS (83.4% vs. 76.1%) and 3-year OS rates (91.6% vs. 85.8%). The same was reported for HR-negative tumors for DFS (82.8% vs. 79.3%), but not for OS [14]. In the present analysis, the pCR rates were also statistically significantly different between HER2-0 and HER2-low-positive tumors (*p* < 0.001)—29.3% in HER2-0 patients and 20.1% in HER2-low-positive patients. No significant differences were observed in the HR-positive or HR-negative subgroups (HR-positive *p* = 0.315, HR-negative *p* = 0.573). The univariate and multivariate analyses showed that patients with HER2-0 tumors have a statistically significantly higher chance for pCR (for example, in the multivariable analysis, ypT0 ypN0: OR 0.611, 95% CI, 0.45 to 0.84, *p* = 0.002), but no differences were seen in the HR-positive or HR-negative subgroups. The effect seen in the overall group therefore seems to be more of a statistical carry-over effect out of the HR-negative patient group than a clinical effect. Differences between these results and Denkert et al. include their period of recruitment, which was between 2012 and 2019, compared to 1998 until now in the present study, and the possible therapeutic effect of study drugs not used in clinical routine work (for example, nab-paclitaxel, denosumab, and trastuzumab s.c.) in comparison with the real-world analysis. This might also influence the OR [14]. Kang and coworkers reported similar results, with a significantly higher pCR rate for HER2-0 patients (14.7% vs. 9.8%, *p* = 0.003) in comparison with HER2-low-positive patients, but no differences in the HR-positive (*p* = 0.4) or HR-negative subgroups (*p* = 0.3) [25]. Another group of researchers also reported results similar to those in the present study. A total of 675 patients with neoadjuvant chemotherapy were analyzed, and those with HER2-0 tumors had significantly higher pCR rates than HER2-low-positive patients (26.6% vs. 16.6%, *p* = 0.002). Similarly, no statistically significant differences in pCR rates between HER2-low-positive and HER2-0 tumors were seen in the HR-positive or HR-negative subgroups [26].

The present study also focused on the long-term follow-up, which showed that the DFS was statistically significantly different (log-rank test, *p* = 0.020) for HR-positive patients (*n* = 897). The same was also found for the OS (log-rank test, *p* = 0.012). In HR-negative patients (*n* = 476), the OS was statistically significantly different (log-rank test, *p* = 0.032), without any statistically significant difference for DFS (log-rank test, *p* = 0.232). Various recent publications have reported on the prognostic significance of HER2-low-positive vs. HER2-0 breast cancers. In a large-scale retrospective study, 109,588 patients received neoadjuvant chemotherapy; 23.6% of the HER2-0 patients and 16.3% of the HER2-low-positive patients who were included experienced pCR, a finding that is consistent with the present results (20.1% to 29.1%), and patients with HER2-low-positive tumors also had a significantly longer overall survival, with minimal effect (HR 0.98; 95% CI, 0.97 to 0.99, *p* < 0.001) [27].

HR-negative patients without a pCR have a much poorer prognosis than patients with a pCR. In a meta-analysis published in 2014, triple-negative and HER2-positive cancers showed strong associations between pCR and long-term survival. In patients with triple-negative tumors, the hazard ratio for OS was 0.16 (0.11–0.25), and for HER2-positive tumors the hazard ratio for OS was 0.08 (0.03–0.22) [17,28]. Other studies, such as the CREATE-X trial (Clinical Trials Registry number: UMIN000000843), have observed significant improvement in long-term outcomes for triple-negative and HR-positive patients without pCR and with positive lymph nodes who received post-neoadjuvant capecitabine in comparison with a control group [29]. The data from the present study show, in addition, that HR-negative, HER2-0 patients have significantly poorer outcomes than those with HER2-low-positive, HR-negative tumors. These truly triple-negative patients still have the poorest prognosis, and additional post-neoadjuvant treatment strategies are needed for this subgroup.

In an analysis of HER2-low-positive breast cancer patients in comparison with HER2-0 patients, Almstedt and coworkers focused on 410 node-negative patients without adjuvant systemic treatment. They also observed a significant difference in DFS for all patients and in OS only for HR-positive patients. Their cohort is different from the present one, since all of the patients in this study were treated with systemic neoadjuvant chemotherapy, but the findings appear to be comparable over a long-term follow-up [15]. The present study focuses on patients with high-risk tumors and on the question of which cohort has an essential need for further evaluations for post-neoadjuvant treatment studies.

A large-scale retrospective analysis including 1,136,016 patients that identified only minimal prognostic differences between HER2-low and HER2-0 breast cancer appears to show that HER2-low-positive tumors are not a new biological class [27]. Nevertheless, it is important to identify a new therapeutic subgroup of breast cancer patients as a HER2-low-positive cohort with prospective treatment chances, and this approach is also supported by the national guidelines in Germany [30]. The retrospective analysis also shows that, as in the present study, HER2-low-positive, HR-negative patients are underrepresented, even though they are an interesting cohort for further treatment strategies, like endocrine-refractory, HR-positive breast cancer patients.

### Limitations

One limitation of this study is the heterogeneity of the patient cohort. A total of 476 HR-negative patients and 897 HR-positive patients were examined, and the interpretation of the OR for pCR rates therefore requires caution. The retrospective and single-center nature of the analysis is a further limitation. Another limitation is the long-term follow-up period, which, despite being one strength of the analysis, means that the therapy for breast cancer patients changed over time, with potentially confounding predictive effects. As discussed above, heterogeneity and interobserver variability are high in HER2-negative tumors and can therefore interfere with the results.

## 5. Conclusions

A new group of breast cancer patients with different survival outcomes can be differentiated immunohistochemically, although at times, this may remain without clinical impact if the patients are classified as HER2-0 or HER2-low-positive and the pathology is sufficiently accurate. There are also new cohorts of patients with different overall survival outcomes and, therefore, different treatment needs.

## Figures and Tables

**Figure 1 cancers-15-04678-f001:**
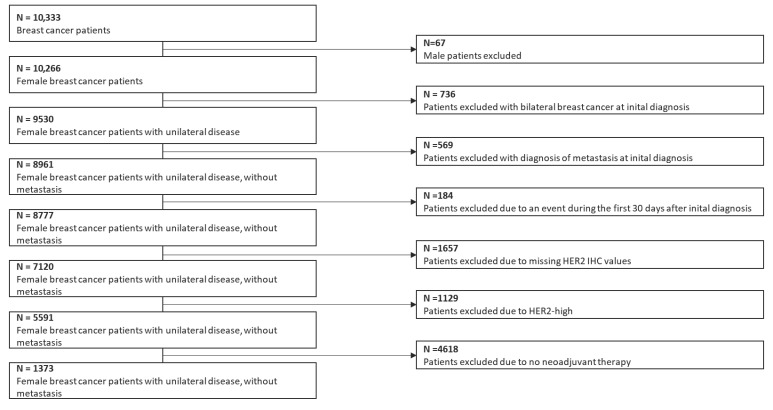
Patient selection and exclusion criteria in this study.

**Figure 2 cancers-15-04678-f002:**
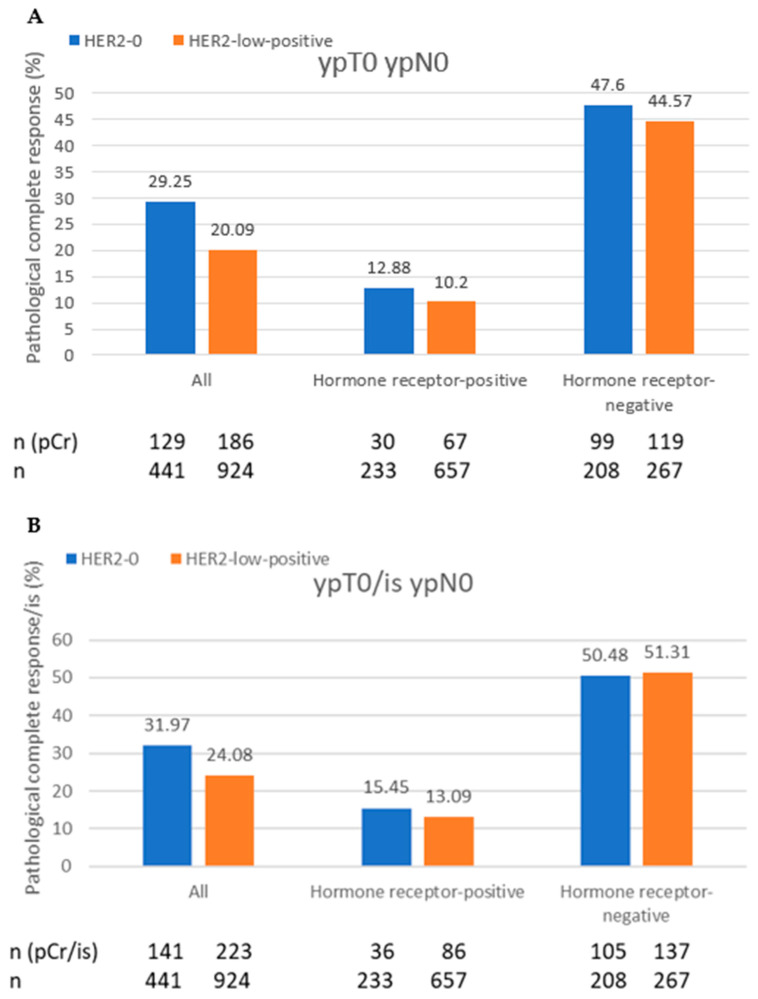
(**A**,**B**) Pathological complete response (pCR) rates in patients with HER2-0 and HER2-low-positive breast cancer. Comparison of pCR rates (**A**) and pCR/is, irrespective of in situ lesions (**B**).

**Figure 3 cancers-15-04678-f003:**
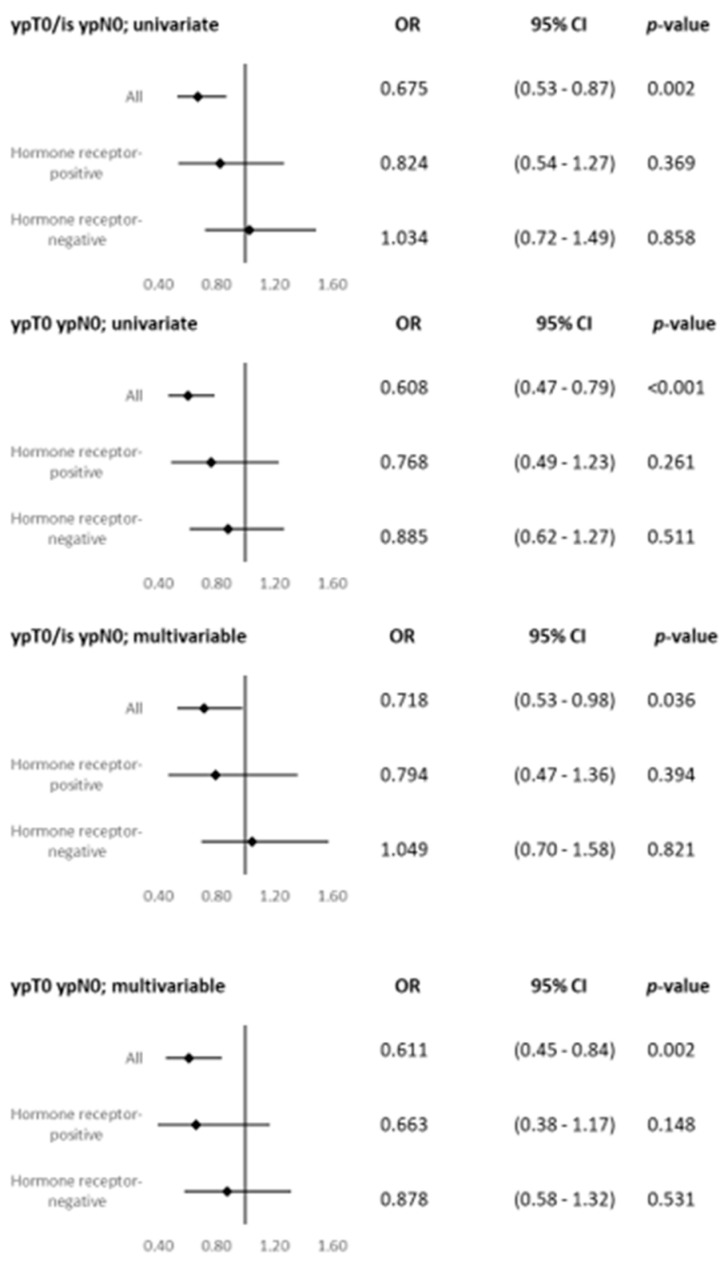
Pathological complete response rate in HER2-0 and HER2-low-positive breast cancer; forest plots for univariate and multivariable logistic regression analysis of pathological complete response rates. CI, confidence intervals; OR, odds ratio; pCR, pathological complete response; pCR/is, pathological complete response, irrespective of in situ lesions.

**Figure 4 cancers-15-04678-f004:**
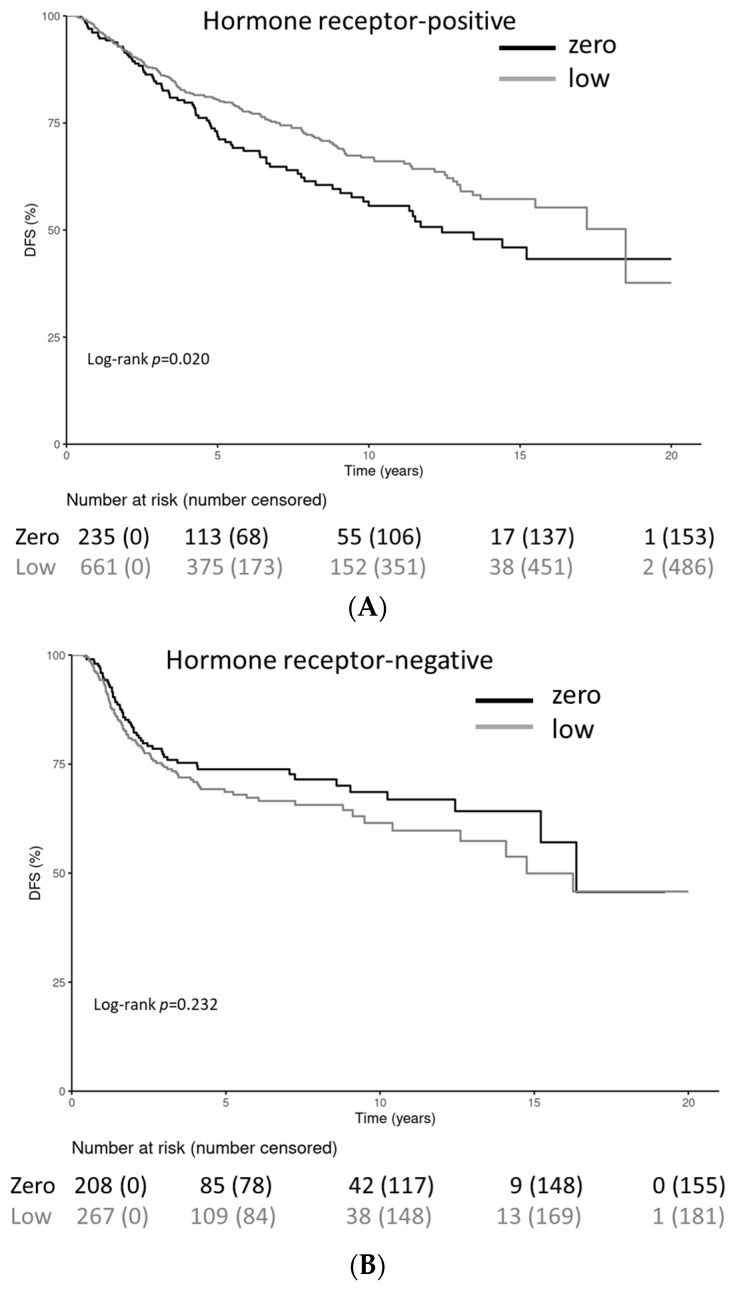
(**A**–**C**). Kaplan–Meier curves for disease-free survival in hormone receptor-negative and hormone receptor-positive patients and the overall cohort with either HER2-0 or HER2-low-positive tumors. Univariate and multivariate Cox regression model for disease-free survival in patients with HER2-0 versus HER2-low-positive, hormone receptor-positive patients. (**A**) The Kaplan–Meier curve for disease-free survival in hormone receptor-negative patients. (**B**) The Kaplan–Meier curve for disease-free survival in hormone receptor-positive patients. (**C**) The Kaplan–Meier curve for disease-free survival in all patients.

**Figure 5 cancers-15-04678-f005:**
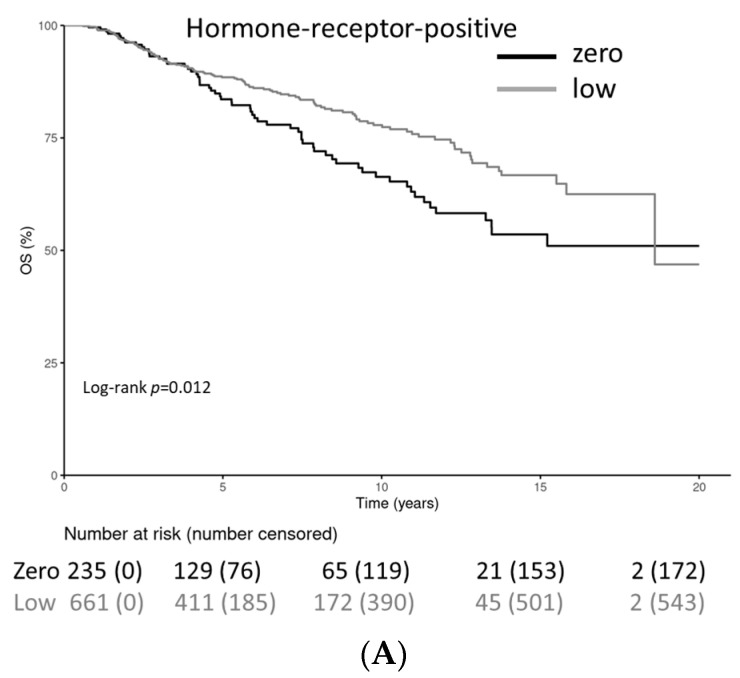
(**A**–**C**) Kaplan–Meier curves for overall survival in hormone receptor-negative and hormone receptor-positive patients and for the overall cohort with either HER2-0 or HER2-low-positive tumors. Univariate and multivariate Cox regression model for overall survival in patients with HER2-0 versus HER2-low-positive, hormone receptor-positive patients. (**A**) The Kaplan–Meier curve for overall survival in hormone receptor-negative patients. (**B**) The Kaplan–Meier curve for overall survival in hormone receptor-positive patients. (**C**) The Kaplan–Meier curve for overall survival in all patients.

**Table 1 cancers-15-04678-t001:** Baseline characteristics for clinical and pathological parameters in 1373 patients in association with HER2 status. Values are given as *n* (%), unless otherwise specified.

Characteristic	HER2-0 (*n* = 443)	HER2-Low (*n* = 930)
Age		
<50	203 (45.8)	375 (40.3)
≥50	240 (54.1)	555 (59.7)
BMI		
Low	14 (3.3)	12 (1.4)
Medium	190 (45.0)	426 (47.8)
High	218 (51.7)	453 (50.8)
Tumor size		
T1	108 (24.6)	226 (24.5)
T2–4	331 (75.4)	698 (75.5)
Nodal status		
N+	209 (47.8)	486 (52.8)
N0	228 (52.2)	435 (47.2)
Histology		
Ductal	302 (68.6)	666 (71.7)
Lobular	46 (10.5)	78 (8.4)
Other	92 (20.9)	185 (19.9)
Grading		
G1	20 (4.6)	32 (3.5)
G2	156 (35.78)	351 (37.9)
G3	260 (59.6)	544 (58.7)
HR		
HR-negative	208 (47.0)	267 (28.8)
HR-positive	235 (53.0)	661 (71.2)
Ki-67		
≤15%	87 (20.4)	196 (21.4)
>15–25%	87 (20.4)	252 (27.5)
>35%	253 (59.3)	469 (51.2)
pCR (ypT0 ypN0)		
Yes	129 (29.3)	186 (20.1)
No	312 (70.8)	740 (79.9)
pCR/is (ypT0/is ypN0)		
Yes	141 (32.0)	223 (24.1)
No	300 (68.0)	703 (76.0)
Surgery Kind		
Breast-conserving surgery	282 (63.6)	599 (64.5)
Mastectomy	161 (36.3)	331 (35.6)
Chemotherapy regimes		
Anthracylcine and taxane	236 (53.3)	542 (58.3)
(+/−platinum)		
Platinum-based and taxane	83 (19.3)	152 (16.3)
(Anthracycline-free)		
CMF	4 (0.01)	25 (0.03)
Platinum monotherapy	0	2 (0.02)
Taxane monotherapy	17 (3.9)	52 (5.6)
Anthracycline monotherapy	56 (13.0)	98 (10.5)
Other	27 (6.3)	35 (3.7)
Missing values	17 (3.9)	24 (2.6)
Adjuvant therapy		
Radiotherapy	278 (73.7)	576 (74.4)
Missing values (radiotherapy)	53 (12.3)	156 (16.8)
Endocrine therapy	165 (44.9)	469 (61.2)
Missing values (endocrine therapy)	54 (12.3)	170 (18.3)

BMI, body mass index; G, grade; HER2, human epidermal growth factor receptor 2; HR, hormone receptor; N, lymph node; pCR, pathological complete response; pCR/is, pathological complete response, irrespective of in situ lesions; T, tumor size; CMF cyclophosphamide, methotrexate, 5-Fluorouracil.

**Table 2 cancers-15-04678-t002:** Chemotherapy regimes for HER2-0 and HER2-low-positive breast cancer patients divided into hormone receptor-positive and negative subgroups, showing pathological complete response rates and pathological complete response, irrespective of in situ lesions rates.

Chemotherapy	HER2-0 HR-Positive pCr; pCr/is (*n* = 235)	HER2-0 HR-Negative pCr; pCr/is (*n* = 208)	HER2-Low HR-Positive pCr; pCr/is (*n* = 663)	HER2-Low HR-Negative pCr; pCr/is (*n* = 267)
Anthracylcine and taxane	15.9; 19.3 (145)	46.2; 50.5 (91)	40.7; 53.8 (451)	40.7; 22.2 (91)
Platinum and taxane	60.0; 60.0 (10)	58.9; 60.3 (73)	54.5; 57.6 (33)	53.9; 62.2 (119)
CMF	0.0; 0.0 (4)	0	9.5; 9.5 (21)	50.0; 50.0 (4)
Platinum monotherapy	0	0	0	50.0; 50.0 (2)
Taxane monotherapy	0	23.5; 23.5 (17)	11.8; 14.7 (34)	50.0; 50.0 (18)
Anthracycline monotherapy	0.0; 0.0 (40)	18.8; 25.0 (16)	6.9; 9.7 (72)	19.2; 23.1 (26)

HER2, human epidermal growth factor receptor 2; HR, hormone receptor; pCR, pathological complete response; pCR/is, pathological complete response, irrespective of in situ lesions; CMF cyclophosphamide, methotrexate, 5-Fluorouracil.

**Table 3 cancers-15-04678-t003:** Multivariate Cox regression model for disease-free survival in HER2-0 versus HER2-low-positive patients, divided into two groups of hormone receptor-positive patients and hormone receptor-negative patients.

Predictor	Multivariate Hazard Ratio for HR-Positive Patients	*p* Value	Multivariate Hazard Ratio for HR-Negative Patients
Age			
<50			
≥50	1.075 (0.815; 1.417)	0.609	0.513 (0.344; 0.765)
BMI			
Low			
Medium	0.407 (0.126; 1.316)	0.133	1.254 (0.448; 3.51)
High	0.443 (0.136; 1.439)	0.175	1.574 (0.558; 4.439)
Tumor size			
T1			
T2	1.333 (0.862; 2.06)	0.196	3.851 (2.135; 6.944)
T3	1.848 (0.968; 3.528)	0.063	6.06 (2.519; 14.576)
T4	2.408 (1.43; 4.057)	0.001	5.101 (2.816; 13.216)
Nodal status			
N0			
N1	1.576 (1.187; 2.093)	0.002	2.139 (1.452; 3.151)
Grading			
G1			
G2	1.013 (0.568; 1.808)	0.964	1.544 (0.319; 7.474)
G3	1.06 (0.563; 1.995)	0.856	1.635 (0.325; 8.217)
HER2 status			
Zero			
Low	0.704 (0.53; 0.934)	0.015	1.554 (1.064; 2.271)
Ki-67			
≤15%			
>15–35%	1.23 (0.881; 1.717)	0.223	0.435 (0.158; 1.202)
>35%	2.04 (1.395; 2.984)	< 0.001	0.407 (0.167; 0.993)
pCR (ypT0 ypN0)			
Yes			
No	4.242 (1.96; 9.184)	< 0.001	3.68 (2.342; 5.783)

BMI, body mass index; G, grade; HER2, human epidermal growth factor receptor 2; HR, hormone receptor; N, lymph node; pCR, pathological complete response; T, tumor size.

**Table 4 cancers-15-04678-t004:** Multivariate Cox regression model for overall survival of HER2-0 versus HER2-low-positive patients, divided into two groups of hormone receptor-positive patients and hormone receptor-negative patients.

Predictor	Multivariate Hazard Ratio for HR-Positive Patients	*p* Value	Multivariate Hazard Ratio for HR-Negative Patients
Age			
<50			
≥50	1.513 (1.075; 2.129)	0.018	0.615 (0.386; 0.978)
BMI			
Low			
Medium	1.897 (0.254; 14.168)	0.533	1.374 (0.416; 4.535)
High	1.68 (0.224; 12.597)	0.614	1.371 (0.411; 4.57)
Tumor size			
T1			
T2	1.209 (0.703; 2.08)	0.493	4.23 (2.011; 8.897)
T3	2.499 (1.208; 5.171)	0.014	6.468 (2.345; 17.843)
T4	2.614 (1.4; 4.878)	0.003	8.926 (3.604; 22.106)
Nodal status			
N0			
N1	1.454 (1.037; 2.04)	0.03	2.414 (1.516; 3.843)
Grading			
G1			
G2	0.665 (0.367; 1.202)	0.177	0.848 (0.159; 4.515)
G3	0.671 (0.343; 1.315)	0.246	0.866 (0.155; 4.844)
HER2 status			
Zero			
Low	0.643 (0.46; 0.898)	0.01	1.906 (1.214; 2.991)
Ki-67			
≤15%			
>15–35%	1.086 (0.726; 1.623)	0.689	0.58 (0.166; 2.021)
>35%	2.339 (1.492; 3.666)	<0.001	0.675 (0.233; 1.952)
pCR (ypT0/ypN0)			
Yes			
No	4.296 (1.716; 10.755)	0.002	3.368 (1.996; 5.682)

BMI, body mass index; G, grade; HER2, human epidermal growth factor receptor 2; HR, hormone receptor; N, lymph node; pCR, pathological complete response; T, tumor size.

## Data Availability

The datasets are available from the corresponding author on reasonable request.

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
