# Peer review of "Clinical Characteristics and Prognosis of HER2-0 and HER2-Low-Positive Breast Cancer Patients: Real-World Data from Patients Treated with Neoadjuvant Chemotherapy"

_cancers, 2023, doi:10.3390/cancers15194678_

Round 1

Reviewer 1 Report

In this study, the authors observed long-term survival outcomes in patients with HER2-0 and HER2-low-positive breast cancer treated with neoadjuvant chemotherapy. The article presents interesting results, the selection of participants in the experiment is logically made, the material is consistently presented. However, inclusion criteria for patients included neoadjuvant chemotherapy, but chemotherapy regimens are not described. Were the treatment regimens different? Drugs? There is also no information about surgical treatment, I believe that without studying this block of information, it is inappropriate to talk about the survival of patients.

Author Response

Dear Editors,

Dear reviewer,

Dear ladies and gentlemen,

We are grateful for your comprehensive review and we appreciate your effort in time and your instructive and helpful comments to improve our manuscript. We have revised our manuscript according to your instructions. We hope you are satisfied with the changes of our manuscript. If further revisions should be requested, we will do so. Please find below our point-for-point replies to your comments.

Yours sincerely,

Patrik Pöschke and Paul Gaß on behalf of the authors

Reviewer 1

In this study, the authors observed long-term survival outcomes in patients with HER2-0 and HER2-low-positive breast cancer treated with neoadjuvant chemotherapy. The article presents interesting results, the selection of participants in the experiment is logically made, the material is consistently presented. However, inclusion criteria for patients included neoadjuvant chemotherapy, but chemotherapy regimens are not described. Were the treatment regimens different? Drugs? There is also no information about surgical treatment, I believe that without studying this block of information, it is inappropriate to talk about the survival of patients.

Reviewer’s comment

Authors’ reply

However, inclusion criteria for patients included neoadjuvant chemotherapy, but chemotherapy regimens are not described. Were the treatment regimens different? Drugs? There is also no information about surgical treatment,

We have included the different chemotherapy regimes, typ of surgery, adjuvant radiotherapy and endocrine therapy in Table 1 „Patient Characteristics“ for HER2-0 and HER2-low-positive patients.

In addition, a new table 2 was introduced to describe in detail what chemotherapy was used for different patient subgroups, divided into HER2-0 and HER2-low-positive and hormone-receptor-positive and negative patients, to show different pCR and pCR/is rates.

Reviewer 2 Report

This study is interesting with clinical significance. The success of DESTINY-Breast04 has brought attention to HER2-0 and HER2 low-positive breast cancer patients. The authors put forward a new and comprehensive point of view on prognosis of HER2-0 and HER2-low-positive breast cancer patients. The followings are some comments to the authors.

Comments:

1. I suggesting providing more baseline information of 1373 patients in Table 1, including neoadjuvant chemotherapy type and cycle, surgical type and systemic treatment after surgical operation. because the factors above play an important role in OS.

2. In figure 2, Patients with HER2-low-positive and HR-positive tumors had significantly longer DFS periods. Whether were the type and cycle of neoadjuvant chemotherapy different between the two groups HER2-0 and HER2 low-positive breast cancer patients with HR-positive?

Author Response

Revision_1 – Repley to reviewer 2

Dear Editors,

Dear reviewer,

Dear ladies and gentlemen,

We are grateful for your comprehensive review and we appreciate your effort in time and your instructive and helpful comments to improve our manuscript. We have revised our manuscript according to your instructions. We hope you are satisfied with the changes of our manuscript. If further revisions should be requested, we will do so. Please find below our point-for-point replies to your comments.

Yours sincerely,

Patrik Pöschke and Paul Gaß on behalf of the authors

Reviewer 2

This study is interesting with clinical significance. The success of DESTINY-Breast04 has brought attention to HER2-0 and HER2 low-positive breast cancer patients. The authors put forward a new and comprehensive point of view on prognosis of HER2-0 and HER2-low-positive breast cancer patients. The followings are some comments to the authors.

Reviewer’s comment

Authors’ reply

1. I suggesting providing more baseline information of 1373 patients in Table 1, including neoadjuvant chemotherapy type and cycle, surgical type and systemic treatment after surgical operation. because the factors above play an important role in OS.

We have included the different chemotherapy regimes, typ of surgery, adjuvant radiotherapy and endocrine therapy in Table 1 „Patient Characteristics“ for HER2-0 and HER2-low-positive patients.

2. In figure 2, Patients with HER2-low-positive and HR-positive tumors had significantly longer DFS periods. Whether were the type and cycle of neoadjuvant chemotherapy different between the two groups HER2-0 and HER2 low-positive breast cancer patients with HR-positive?

In addition, a new table 2 was introduced to describe in detail what chemotherapy was used for different patient subgroups, divided into HER2-0 and HER2-low-positive and hormone-receptor-positive and negative patients, to show different pCR and pCR/is rates.

Round 2

Reviewer 1 Report

I have no further comments on the article. I believe that in its present form the article can be recommended for publication.